# NOVAS: NON-CONVEX OPTIMIZATION VIA ADAPTIVE STOCHASTIC SEARCH FOR END-TO-END LEARNING AND CONTROL

**Ioannis Exarchos**[*]
Department of Computer Science
Stanford University
Stanford, CA 94305
exarchos@stanford.edu

**Marcus A. Pereira**,[*] **Ziyi Wang & Evangelos A. Theodorou**
School of Aerospace Engineering
Georgia Institute of Technology
Atlanta, GA 30332
{marcus.pereira, ziyiwang,
evangelos.theodorou}@gatech.edu

## ABSTRACT

In this work we propose the use of adaptive stochastic search as a building block for general, non-convex optimization operations within deep neural network architectures. Specifically, for an objective function located at some layer in the network and parameterized by some network parameters, we employ adaptive stochastic search to perform optimization over its output. This operation is differentiable and does not obstruct the passing of gradients during backpropagation, thus enabling us to incorporate it as a component in end-to-end learning. We study the proposed optimization module's properties and benchmark it against two existing alternatives on a synthetic energy-based structured prediction task, and further showcase its use in stochastic optimal control applications.

## 1 INTRODUCTION

Deep learning has experienced a drastic increase in the diversity of neural network architectures, both in terms of proposed structure, as well as in the repertoire of operations that define the inter-dependencies of its elements. With respect to the latter, a significant amount of attention has been devoted to incorporating *optimization blocks or modules* operating at some part of the network. This has been motivated by large number of applications, including meta-learning (Finn et al., 2017; Rusu et al., 2018; Bartunov et al., 2019), differentiable physics simulators (de Avila Belbute-Peres et al., 2018), classification (Amos et al., 2019), GANs (Metz et al., 2016), reinforcement learning with constraints, latent spaces, or safety (Amos & Kolter, 2017; Srinivas et al., 2018; Amos & Yarats, 2019; Cheng et al., 2019; Pereira et al., 2020), model predictive control (Amos et al., 2018; Pereira et al., 2018), as well as tasks relying on the use of energy networks (Belanger et al., 2017; Bartunov et al., 2019), among many others. Local[1] optimization modules lead to *nested optimization* operations, as they interact with the global, end-to-end training of the network that contains them. Consider some component within the neural network architecture, e.g. a single layer, whose input and output are $x_i \in \mathbb{R}^n$ and $x_{i+1} \in \mathbb{R}^m$, respectively. Within that layer, the input and output are linked via the solution of the following optimization problem:

$$x_{i+1} = \arg\min_x F(x; x_i, \theta), \tag{1}$$

that is, the output $x_{i+1}$ is defined as the solution to an optimization problem for which the input $x_i$ remains temporarily fixed, i.e., acts as a parameter. Here, $F(x; x_i, \theta) : \mathbb{R}^m \times \mathbb{R}^n \times \Theta \to \mathbb{R}$ is a function possibly further parameterized by some subset of the neural network parameters $\theta \in \Theta$. Note that $x$ here is an independent variable which is free to vary. The result of this optimization could potentially also be subject to a set of (input-dependent) constraints, though in this paper we

---

[*]Equal contribution.

[1]To distinguish between the optimization of the entire network as opposed to that of the optimization module, we frequently refer to the former as global or *outer-loop* optimization and to the latter as local or *inner-loop* optimization.

will consider only unconstrained optimization. It is also important to note that, depending on the problem, $F$ can be a given function, or it can itself be represented by a multi-layer neural network (trained by the outer loop), in which case the aforementioned optimization layer consists of multiple sub-layers and is more accurately described as a *module* rather than a single layer. Examples of this type of optimization are structured prediction energy networks (e.g. Belanger et al. (2017)); another such example is Amos & Kolter (2017) which treats the case of convex $F(\cdot; x_i, \theta)$.

In order to facilitate end-to-end learning over the entire network, the gradient of its loss function $\mathcal{L}$ with respect to $\theta$ will require during backpropagation passing the gradient of the module's output $x^{i+1}$ with respect to parameters $\theta$ and $x_i$. Depending on the nature of the optimization problem under consideration, several procedures have been suggested; among them, particularly appealing is the case of convex optimization (Gould et al., 2016; Johnson et al., 2016; Amos et al., 2017; Amos & Kolter, 2017), in which the aforementioned gradients can be computed efficiently through an application of the implicit function theorem to a set of optimality conditions, such as the KKT conditions. In the case of non-convex functions however, obtaining such gradients is not as straight-forward; solutions involve either forming and solving a locally convex approximation of the problem, or unrolling gradient descent (Domke, 2012; Metz et al., 2016; Belanger et al., 2017; Finn et al., 2017; Srinivas et al., 2018; Rusu et al., 2018; Foerster et al., 2018; Amos et al., 2018). Unrolling gradient descent approximates the $\arg\min$ operator with a fixed number of gradient descent iterations during the forward pass and interprets these as an unrolled compute graph that can be differentiated through during the backward pass. One drawback in using this unrolled gradient descent operation however is the fact that doing so can lead to over-fitting to the selected gradient descent hyper-parameters, such as learning rate and number of iterations. Recently, Amos & Yarats (2019) demonstrated promising results in alleviating this phenomenon by replacing these iterations of gradient descent by iterations of sampling-based optimization, in particular a differentiable approximation of the cross-entropy method. While still unrolling the graph created by the fixed number of iterations, they showed empirically that no over-fitting to the hyper-parameters occurred by performing inference on the trained network with altered inner-loop optimization hyper-parameters. Another significant bottleneck in all methods involving graph unrolling is the number of iterations, which has to be kept low to prevent a prohibitively large graph during backprop, to avoid issues in training.

Note that in eq. (1) the variable of optimization is free to vary independently of the network. This is in contrast to many applications involving nested optimization, mainly in the field of meta-learning, in which the inner loop, rather than optimizing a free variable, performs *adaptation* to an initial value which is supplied to the inner loop by the outer part of the network. For example, MAML (Finn et al., 2017) performs the inner-loop adaptation $\theta \to \theta'$, in which the starting point $\theta$ is not arbitrary (as $x$ is in eq. (1)) but is supplied by the network. Thus, in the context of *adaptation*, unrolling the inner-loop graph during back-prop is generally necessary to trace the adaptation back to the particular network-supplied initial value. Two notable exceptions are first-order MAML (Finn et al., 2017; Nichol et al., 2018), which ignores second derivative terms, and implicit MAML (Rajeswaran et al., 2019), which relies on local curvature estimation.

In this paper we propose Non-convex Optimization Via Adaptive Stochastic Search (NOVAS), a module for differentiable, non-convex optimization. The backbone of this module is adaptive stochastic search (Zhou & Hu, 2014), a sampling-based method within the field of stochastic optimization. The contributions of our work are as follows: (A). We demonstrate that the NOVAS module does not over-fit to optimization hyper-parameters and offers improved speed and convergence rate over its alternative (Amos & Yarats, 2019). (B). If the inner-loop variable of optimization is free to vary (i.e., the problem fits the definition given by eq. (1)), *we show that there is no need to unroll the graph during the back-propagation of gradients*. The latter advantage is critical, as it drastically reduces the size of the overall end-to-end computation graph, thus facilitating improved ability to learn with higher convergence rates, improved speed, and reduced memory requirements. Furthermore, it allows us to use a higher number of inner-loop iterations. (C). If the inner-loop represents an adaptation to a network-supplied value as it is the case in meta-learning applications, NOVAS may still be used in lieu of the gradient descent rule (though unrolling the graph may be necessary here). Testing NOVAS in such a setting is left for future work. (D). We combine the NOVAS module with the framework of *deep FBSDEs*, a neural network-based approach to solving nonlinear partial differential equations (PDEs). This combination allows us to solve Hamilton-Jacobi-Bellman (HJB) PDEs of the most general form, i.e., those in which the $\min$ operator does not have a closed-form solution, a class of problems that was previously impossible to address due to the non-convexity of

the corresponding Hamiltonian. We validate the algorithm on a cart-pole task and demonstrate its scalability on a 101-dimensional continuous-time portfolio selection problem. The code is available at `https://github.com/iexarchos/NOVAS.git`

## 2 FURTHER BACKGROUND AND RELATED WORK

**Relation to Differentiable Cross-Entropy:** Particular importance should be given to Amos & Yarats (2019), since, to the best of our knowledge, it is the first to suggest sampling-based optimization instead of gradient descent, and features some similarities with our approach. The authors therein propose a differentiable approximation of the cross-entropy method (CEM) (Rubinstein, 2001; De Boer et al., 2005), called differentiable cross-entropy (DCEM). To obtain this approximation, they need to approximate CEM's eliteness threshold operation, which is non-differentiable. This is done by solving an additional, convex optimization problem separately for each inner loop step (and separately for each sample of $x_i$ in the batch, resulting in a total of $N \times M \times K$ *additional convex optimization problems*, with $N$: batch size, $M$: number of inner loop iterations, $K$: number of outer loop iterations, i.e. training epochs). After CEM has been locally approximated by DCEM, they replace the usual inner-loop gradient descent steps with DCEM steps, and the entire inner-loop optimization graph is unrolled during the backward pass. Our method differs from this approach in the following ways: 1. we employ the already differentiable adaptive stochastic search algorithm, thus not having to solve any additional optimization problem to obtain a differentiable approximation (speed improvement), while also showing some convergence rate improvements, and most importantly 2. In the case of inner-loop optimization over an independent variable (e.g., such as the problem defined by eq. (1)), we *do not unroll the optimization graph, but instead pass the gradients only through the last inner-loop iteration*. This drastically reduces its size during backpropagation, increasing speed, reducing memory requirements, and facilitating easier learning.

**Sampling-based Optimization:** Adaptive stochastic search (Zhou & Hu, 2014) is a sampling-based method within stochastic optimization that transforms the original optimization problem via a probabilistic approximation. The core concept behind this algorithm is approximating the gradient of the objective function by evaluating random perturbations around some nominal value of the independent variable, a concept that also appears under the name Stochastic Variational Optimization and shares many similarities with natural evolution strategies (Bird et al., 2018). Another comparable approach is CEM (Rubinstein, 2001; De Boer et al., 2005). In contrast to adaptive stochastic search, CEM is non-differentiable (due to the eliteness threshold) and the parameters are typically updated *de novo* in each iteration, rather than as a gradient descent update to the parameter values of the previous iteration. In the case of Gaussian distributions, the difference between CEM and adaptive stochastic search boils down to the following: in adaptive stochastic search, the mean gets updated by calculating the average of all sampled variable values *weighted* by a typically exponential mapping of their corresponding objective function values, whereas in CEM only the top-$k$ performing values are used, and are weighted equally. Furthermore, this difference can be made even smaller if one replaces the exponential mapping in the former method with a differentiable (sigmoid) function that approximates the eliteness operation. More details are available in the Appendix.

**Deep Learning Approaches for PDEs and FBSDEs:** There has been a recent surge in research and literature in applying deep learning to approximate solutions of high-dimensional PDEs. The transition from a PDE formulation to a trainable neural network is done via the concept of a system of Forward-Backward Stochastic Differential Equations (FBSDEs). Specifically, certain PDE solutions are linked to solutions of FBSDEs. Systems of FBSDEs can be interpreted as a stochastic equivalent to a two-point boundary value problem, and can be solved using a suitably defined deep neural network architecture. This is known in the literature as the *deep FBSDE* approach (Han et al., 2018; Raissi, 2018). While applied in high-dimensional PDEs, the aforementioned results have seen very limited applicability in the field of optimal control. Indeed, the HJB PDE in control theory has a much more complicated structure, and in its general form involves a min operator applied on its Hamiltonian term over the control input. Exploiting certain structures of system dynamics and cost functions that allowed for a closed-form expression for this operator, Exarchos & Theodorou (2018); Exarchos et al. (2018; 2019) developed a framework for control using FBSDEs, which was then translated to a deep neural network setting in Pereira et al. (2019b); Wang et al. (2019). *In this work, we incorporate the NOVAS module inside deep FBSDE neural network architectures to account for PDEs lacking a closed-form expression for their* min *and/or* max *operators.* Thus, we are able to

address the most general description of a HJB PDE in which the corresponding Hamiltonian is non-convex. More information concerning the deep FBSDE framework can be found in the Appendix.

## 3 NON-CONVEX OPTIMIZATION VIA ADAPTIVE STOCHASTIC SEARCH

The cornerstone of our approach is a method within stochastic optimization called *adaptive stochastic search* (Zhou & Hu, 2014). Adaptive stochastic search addresses the general maximization[2] problem

$$x^* \in \arg\max_{x \in \mathcal{X}} F(x), \qquad \mathcal{X} \subseteq \mathbb{R}^n, \tag{2}$$

with $\mathcal{X}$ being non-empty and compact, and $F : \mathcal{X} \to \mathbb{R}$ a real-valued, non-convex, potentially discontinuous and non-differentiable function. Instead of dealing with this function that lacks desirable properties such as smoothness, adaptive stochastic search proposes the solution of a stochastic approximation of this problem in which $x$ is drawn from a selected probability distribution $f(x; \rho)$ of the exponential family with parameters $\rho$ and solve

$$\rho^* = \arg\max_{\rho} \int F(x) f(x; \rho) \mathrm{d}x = \mathbb{E}_{\rho} \left[ F(x) \right].$$

This new objective function, due to its probabilistic nature, exhibits desirable properties for optimization. Algorithmically, this can be facilitated by introducing a natural log and a shape function $S(\cdot) : \mathbb{R} \to \mathbb{R}^+$ which is continuous, non-decreasing, and with a non-negative lower bound (an example of such a function would be the exponential). Due to their properties, passing $F(x)$ through $S(\cdot)$ and the log does not affect the optimal solution. The final optimization problem is then

$$\rho^* = \arg\max_{\rho} \ln \int S(F(x)) f(x; \rho) \mathrm{d}x = \ln \mathbb{E}_{\rho} \left[ S(F(x)) \right]. \tag{3}$$

To address this optimization problem, one can sample candidate solutions $x$ from $f(x; \rho)$ in the solution space $\mathcal{X}$, and then use a gradient ascent method on eq. (3) to update the parameter $\rho$. Depending on the chosen probability distribution for sampling $x$, a closed-form solution for the gradient of the above objective function with respect to $\rho$ is available. Thus, while still being a sampling-based method at its core, adaptive stochastic search employs gradient ascent on a *probabilistic mapping* of the initial objective function. While any probability density function of the exponential family will work, in our work we sample $x$ from a Gaussian distribution. The resulting update scheme is shown in Alg. 1. More details concerning its derivation are included in the Appendix.

As described in the introduction, the standard approach employed in the literature for non-convex inner-loop optimization is to apply an optimization procedure (either gradient descent or DCEM) for a fixed number of iterations during the forward pass and interpret these as an unrolled compute graph that can be differentiated through during the backward pass. In this work we argue that this needs to be done only in cases of *adaptation* (e.g., in meta-learning): the variable to be adapted is supplied to the inner-loop by the outer part of the network, and is adapted using a rule such as gradient descent with respect to the inner-loop objective function for a fixed number of steps. Crucially, the process is not initialized with an arbitrary initial value for the adapted variable but with the one that is supplied by the network; the backward pass which needs to pass through the optimization module also needs to flow through the input of the variable of optimization in the module. Thus, the variable's values pre- and post-adaptation need to be linked via the unrolled computational graph of the adaptation. In contrast, in the case in which the variable of optimization is not an input to the layer, but rather is allowed to vary freely and independently of the outer part of the network (e.g. as described by problem (1)), such a process is not only unnecessary, but further leads to complications such as reduced network trainability and learning speed, as well as increased memory usage and computation time. Given that in this case the variable of optimization is initialized by what is typically no more than a random guess, there is no need to trace the gradients during backpropagation all the way back to that random guess.

After fixing a number of iterations for optimization, say, $n$, a simple way to implement NOVAS in a non-unrolled fashion is to take $n - 1$ of the iterations off the graph and perform only the $n$-th on

---

[2] While presented for maximization, we deploy it for minimization by switching the sign of the objective function.

---

**Algorithm 1:** Non-convex Optimization Via Adaptive Stochastic Search (NOVAS)

---

**Given:**
Neural network architecture containing NOVAS module, mini-batch dataset $\{X_j, Y_j\}_{j=1}^{J}$,
NOVAS objective function $F(\cdot; x_i, \theta)$.
**Parameters:**
Neural network parameters: number of layers $L$, layer transformations $f_i$, network parameters $\theta$.
NOVAS parameters: initial mean and standard deviation $\mu_0, \sigma_0$, learning rate $\alpha$, shape function
$S$, number of samples M, number of iterations $N$, small positive number $\varepsilon = 10^{-3}$ (sampling
variance lower bound).
**Set** $x_1 \leftarrow \{X_j\}$
**for** $i = 1$ **to** $L$ **do**
    **if** $f_i$ is NOVAS layer **then**
        **Set** $\mu \leftarrow \mu_0$, $\sigma \leftarrow \sigma_0$
        **for** $n = 1$ **to** $N - 1$ (***off-graph operations***) **do**
            $(\mu, \sigma) \leftarrow$ **NOVAS_Layer** $(\mu, \sigma, \alpha, S, M, N, \epsilon, F)$
        **end for**
        $(\mu, \sigma) \leftarrow$ **NOVAS_Layer** $(\mu, \sigma, \alpha, S, M, N, \epsilon, F)$
        $x_{i+1} = \mu$
    **else**
        $x_{i+1} = f_i(x_i; \theta)$
    **end if**
    $\{\hat{Y}_j\} = x_{L+1}$
    **Compute Loss:** $\mathcal{L} = \text{MSE}(Y_j, \hat{Y}_j)$
    **Update Parameters:** $\theta \leftarrow \text{Adam}(\mathcal{L}, \theta)$
**end for**

---

$(\mu, \sigma) \leftarrow$ **NOVAS_Layer** $(\mu, \sigma, \alpha, S, M, N, \epsilon, F)$

---

Generate $M$ samples of $x^m \sim \mathcal{N}(\mu, \sigma^2)$,      $m = 1, \ldots, M$;
**for** $m = 1$ **to** $M$ (***vectorized operation***) **do**
    Evaluate $F^m = F(x^m)$ for maximization or $F^m = -F(x^m)$ for minimization;
    Normalize $F^m = \big(F^m - \min_m(F^m)\big)/\big(\max_m(F^m) - \min_m(F^m)\big)$;
    Apply shape function $S^m = S(F^m)$;
    Normalize $S^m = S^m / \sum_{m=1}^{M} S^m$;
**end for**
Update $\mu = \mu + \alpha \sum_{m=1}^{M} S^m(x^m - \mu)$,      $\sigma = \text{sqrt}(\sum_{m=1}^{M} S^m(x^m - \mu)^2 + \varepsilon)$;

---

graph. This amounts essentially to getting a good initial guess solution and performing a single step
optimization. The latter is enough to supply gradient information during back-prop as it relates to the
relevant, optimized value $x^*$, rather than the intermediate steps forming the trajectory from its initial
guess to its final optimal value. From a coding perspective, most automatic differentiation packages
allow for localized deactivation of gradient information flow; in PyTorch (Paszke et al., 2019), this
is as simple as adding a "`with torch.no_grad():`" for the $n - 1$ first iterations. With regards
to the proposed Alg. 1, we would like to mention following: A. The forward pass of the given neural
network containing the NOVAS module is a nested operation of layer transformations $f_i$ given by
$Y_j = f_L(f_{L-1}(f_{L-2}(\ldots f_1(X_j) \ldots)))$, wherein $f_i$ can be either a **NOVAS_layer** or any standard
neural network layer. Also, note that the above transformation applies to recurrent layers with the
forward pass unrolled wherein each $f_i$ corresponds to a time step. B. For clarity of presentation,
we outline the forward and backward (i.e. backprop) passes for a single sampled mini-batch from
the training dataset. The user is free to choose a training strategy that best suits the problem and
apply the proposed algorithm to every sampled mini-batch with any variant of stochastic gradient
descent for the outer loop. C. The "Normalize $F$" operation in **NOVAS_layer** is optional, but may
lead to some numerical improvements. Further algorithm implementation details are given in the
Appendix. Finally, an important remark is necessary concerning the differentiability of the module.
Nonconvex optimization problems do not necessarily have a unique optimum, and as a result, the
$\arg\min$ operator is a set-valued map and not differentiable in the classical sense; even when the

optimum is unique almost everywhere in the parameter space it is possible for the optimal value to be discontinuous in the parameter. For non-global optimizers, an initialization that avoids wrong local optima is crucial. With respect to the latter, there is some evidence (see, e.g., Bharadhwaj et al. (2020)) that sampling-based optimization methods are more robust compared to gradient descent as they evaluate the objective function over an extended area of the input space and, given enough sampling variability for exploration, have thus more chances of escaping a narrow local optimum.

## 4 APPLICATIONS

In this section we explore the properties of NOVAS and test its applicability in a few problems involving end-to-end learning. The first task is a Structured Prediction Energy Network (SPEN) learning task, which we adopted directly from Amos & Yarats (2019). We found it to be an ideal environment to test NOVAS against unrolled DCEM and unrolled gradient descent because it is simple, allows for fast training, and, being two-dimensional, one can visualize the results. We would like to stress that this is merely an example for illustrating various algorithm differences and behavior rather than a claim on state-of-the-art results in the domain of SPENs. We then address two optimal control problems by incorporating the NOVAS module in a deep FBSDE neural network architecture as shown in Fig. 3: the first problem is the cart-pole swing-up task, a low-dimensional problem that has been successfully addressed with already existing deep FBSDE approaches (Pereira et al., 2019b) that exploit the structure of dynamics and cost (dynamics are affine in control and the cost is quadratic in the control) in order to perform minimization of the Hamiltonian explicitly. This problem merely serves as a means to validate the NOVAS-FBSDE algorithm. The second problem demonstrates the establishment of a new state-of-the-art in solving high-dimensional HJB PDEs using the deep FBSDE method; specifically, we address a continuous-time portfolio optimization problem that leads to a general (i.e., without an explicit solution for the $\min$ operator) HJB PDE in 101 dimensions. This HJB PDE form could not be addressed by deep FBSDE methods previously.

### 4.1 STRUCTURED PREDICTION ENERGY NETWORKS

The goal in energy-based learning is to estimate a conditional probability $\mathbb{P}(y|x)$ of an output $y \in \mathcal{Y}$ given an input $x \in \mathcal{X}$ using a parameterized energy function $E(x, y; \theta) : \mathcal{X} \times \mathcal{Y} \times \Theta \to \mathbb{R}$, wherein $\theta \in \Theta$ are the energy function's trainable parameters. The conditional probability is approximated as $\mathbb{P}(y|x) \propto \exp(-E(y; x, \theta))$. Predictions can be made by minimizing the trained energy function with respect to $y$:

$$\hat{y} = \arg\min_{y} E(x, y; \theta). \tag{4}$$

Initially studied in the context of linear energy functions (Taskar et al., 2005; LeCun et al., 2006), the field recently adopted deep neural networks called structured prediction energy networks (SPENs) (Belanger & McCallum, 2016) to increase the complexity of learned energy functions. In particular, Belanger et al. (2017) suggested training SPENs in a supervised manner; unrolled gradient descent is used to obtain $\hat{y}$ which then compared to the ground-truth $y^*$ by a suitable loss function. Mimicking the unrolled gradient descent suggested by Belanger et al. (2017), Amos & Yarats (2019) replaced the gradient descent operations with differentiable cross entropy iterations, also using an unrolled computation graph during backpropagation. Here we adopt the same example as in Amos & Yarats (2019) to benchmark NOVAS against unrolled gradient descent and unrolled DCEM, compare their properties, and further show that unrolling the inner-loop graph is not necessary. We consider the simple regression task where ground-truth data are generated from $f(x) = x \sin(x)$, $x \in [0, 2\pi]$, and we use a neural network of 4 hidden layers to approximate $E$. This problem belongs to the class of problems described by eq. (1) in the introduction: the variable of optimization, $y$, is not an input to the optimization module from the exterior part of the network. The energy function represented by the multi-layer neural network defines the objective function *within* the module (that is, it corresponds to $F(\cdot; x_i, \theta)$ of eq. (1)). While in this case the entire network is within the module, the input could instead be features extracted from an anterior part of the network, e.g. through convolution. The results are shown in Figs. 1 and 2. As can be seen from Fig. 2(a), unrolled gradient descent converges to a very low loss value (which implies good regression performance), but the trained energy function does not reflect the ground-truth relationship, Fig. 1(a). This implies a "faulty" inner-loop optimization, which the energy network itself learns to compensate for. The result resembles more ordinary regression than energy-based structured prediction, since no useful

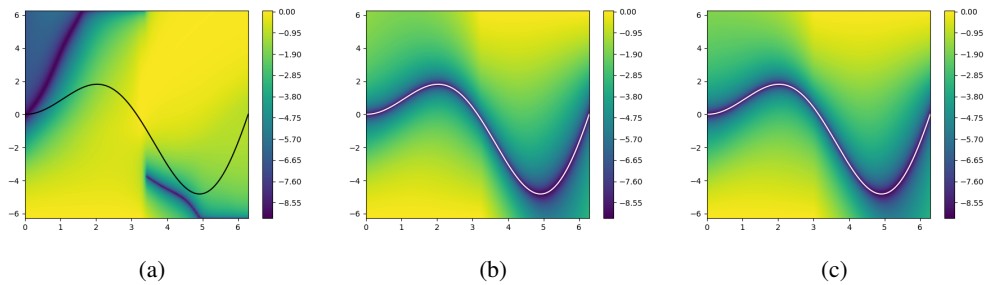

Figure 1: **Learned energy functions:** plots depict the log of the normalized energy function trained using 10 inner loop iterations of (a) unrolled gradient descent (reproduced from Amos & Yarats (2019)), (b) NOVAS with the entire inner-loop graph unrolled, and (c) NOVAS without unrolling (only the last iteration on-graph). Black/white curve denotes the ground truth solution $x \sin(x)$.

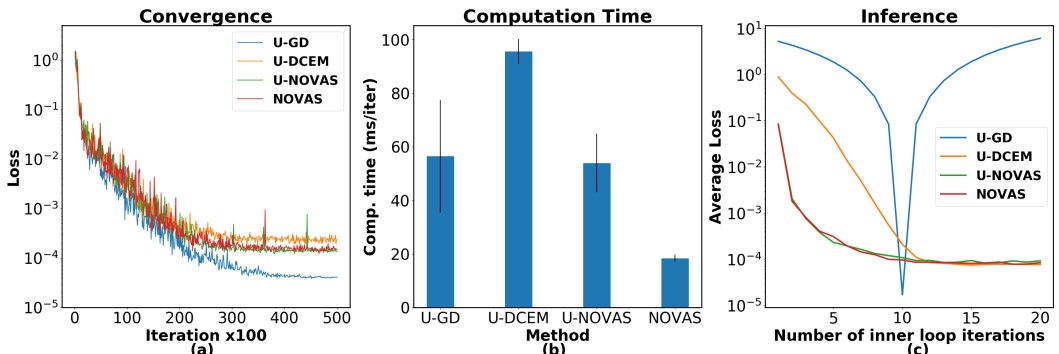

Figure 2: **Convergence analysis and computation time:** (a). convergence (test set loss) and (b). computation time for unrolled gradient descent (U-GD), unrolled DCEM (U-DCEM), NOVAS in unrolled mode (U-NOVAS) and without unrolling (NOVAS). In terms of speed, NOVAS offers an approximate speedup of 5x compared to U-DCEM, and if unrolled is on-par with U-GD. **(c). Inference on trained models with altered inner-loop parameters:** All models were trained using 10 inner loop iterations. Unrolled GD causes the energy network to over-fit to that number (as noted in Amos & Yarats (2019)). Unrolled DCEM does not suffer from this phenomenon, but optimization seems to be less efficient (more inner-loop iterations are required for the same loss as NOVAS). NOVAS, in both its regular and unrolled form does not over-fit, and is the most efficient.

structure is learned; furthermore, changing the inner-loop optimization parameters during inference (after training) leads to an operation that the energy network has not learned to compensate for, as seen in Fig. 2(c). The sampling-based methods of unrolled DCEM and NOVAS both alleviate this phenomenon by learning the correct energy landscape. Furthermore, as seen in Fig. 1(b) and (c), unrolling the graph is unnecessary, and avoiding it leads to a significant speed-up, 5x with respect to unrolled DCEM, Fig. 2(b). Interestingly, an additional benefit is that NOVAS seems to offer a greater inner-loop convergence rate than DCEM (Fig. 2(c)). Due to the simplicity of this example, there is no learning inhibition when using an unrolled graph, as seen from the comparison between NOVAS and unrolled NOVAS. However, this can be the case in more complex tasks and network architectures, as we shall see in Section 4.2. Further details are given in the Appendix.

## 4.2 CONTROL USING FBSDES

### 4.2.1 CART-POLE SWING-UP TASK

We first validate the NOVAS-FBSDE algorithm (neural network architecture seen in Fig. 3) by solving a task whose special structure (dynamics affine in control and cost function quadratic in control) allows for a closed-form solution for the min operator of the Hamiltonian. Because of this special structure, this task can be solved by already existing deep FBSDE approaches (e.g., Pereira et al. (2019b)). Here, we replace the minimization explicit solution with NOVAS. The results are shown in Fig. 4, and are in accordance with results obtained using explicit minimization (see Fig. 6 in Pereira et al. (2019b)). Equations and implementation details are given in the Appendix.

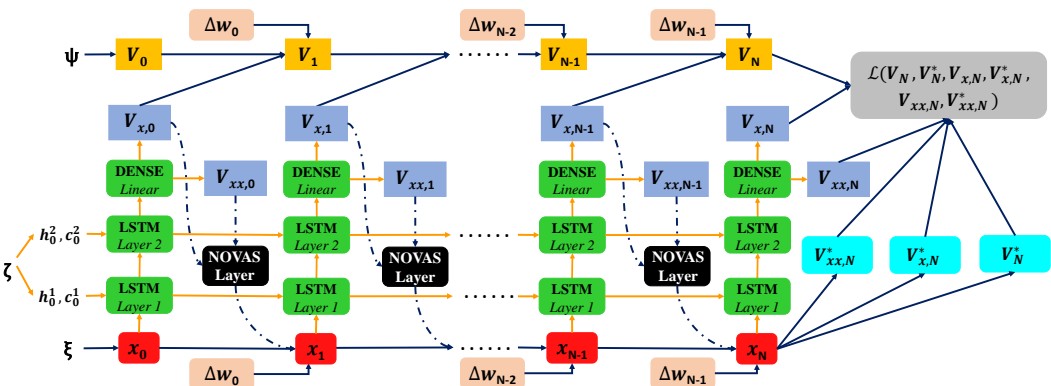

Figure 3: **NOVAS-FBSDE neural network architecture:** the NOVAS module is incorporated in the neural network to minimize the Hamiltonian over the control at each time step.

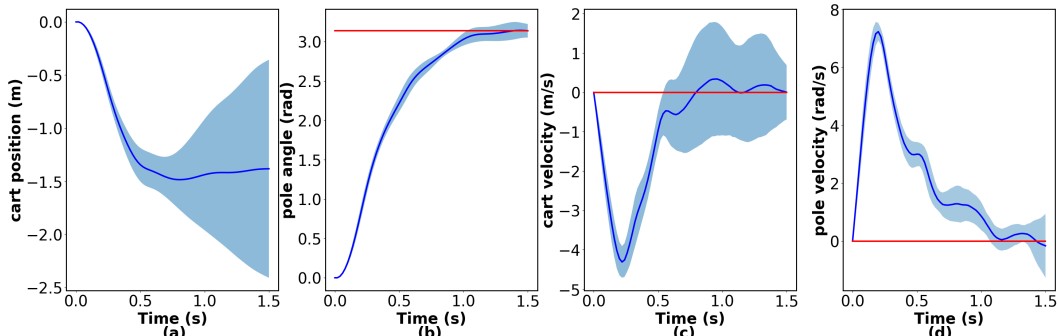

Figure 4: **Cartpole:** Validation of the NOVAS-FBSDE algorithm on a task that allows explicit minimization of the Hamiltonian with respect to the control. Replacing the explicit minimization with NOVAS leads to the same solution (compare with Fig. 6 in Pereira et al. (2019b)). (a) cart position, (b) pole angle, (c) cart velocity (d) pole angular velocity. Blue line denotes mean trajectory, shaded regions show controller's response to injected noise. Target states are indicated with red. Plots above show statistics over 128 test trials.

### 4.2.2 HIGH-DIMENSIONAL, CONTINUOUS-TIME PORTFOLIO SELECTION

We now demonstrate that augmenting the deep FBSDE method with NOVAS allows us to solve general, high-dimensional HJB PDEs by employing NOVAS-FBSDE on a stock portfolio optimization problem, defined as follows: we consider a market index $I$ that consists of $N = 100$ stocks, and select a subset of $M = 20$ of those for trading. There is also a risk-less asset with (relatively low) return rate. We may invest an initial wealth capital $W$ among these $20 + 1$ assets, and the goal is to control the percentage allocation among these assets *over time* such that the wealth outperforms the market index *in probability*. This optimal control formulation leads to a HJB PDE on a state space of $100 + 1$ dimensions (100 stocks of the market plus the wealth process, which incorporates the risk-less asset dynamics. Volatility clearly dominates in such short-term horizons, so a successful trading strategy would be one that increases the odds of beating the market average compared to a random selection. *Index-tracking* and *wealth-maximization* have long been the subject of study from a controls perspective (Primbs, 2007; Bae et al., 2020), though with limited results due to the difficulty in dealing with such high uncertainties. Primbs (2007) investigates a low-dimensional variant of this problem (5 stocks, 3 traded) but does not enforce the constraint that allocation can only be positive (thus leading to negative investments, i.e. borrowing money from stocks), and *tracks* the index (thus paying a penalty also when the portfolio outperforms the index) due to the approach being restricted to consider only quadratic cost functions. We avoid these and enforce positive investments only by applying a `softmax` on the control input, and use $\left(\texttt{softplus}(I - W)\right)^2$ as cost function to incentivize outperforming rather than tracking the index. We consider two alternative investment strategies as baselines: a constant and equal allocation among all traded assets, as well as random allocations. The results are shown in Fig. 5. As indicated by the violin plots, the FBSDE investment strategy outperforms in probability the two alternative strategies, and is the only one who

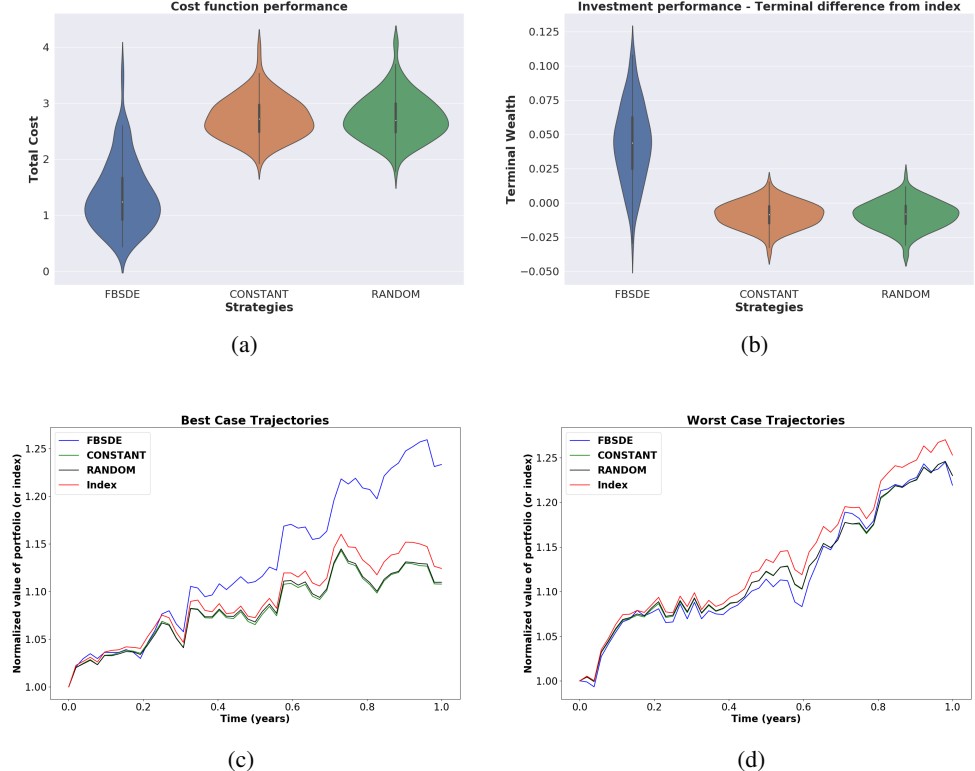

Figure 5: **Portfolio Optimization:** (a) Violin plots for the comparison of trading strategies in terms of the defined loss. The FBSDE controller outperforms in probability both strategies of equal and random allocation among traded stocks. (b) Wealth - market index difference ($W - I$) at the end of one year for the three investment strategies. (c) Trajectory corresponding to the lowest cost (best case). (d) Trajectory corresponding to the highest cost (worst case).

outperforms the market index (by almost 5% on average) at the end of the planning horizon of one year. These statistics are obtained by running 128 market realizations with *test set* volatility profiles (noise profiles not seen during training or validation). All equations and implementation details are provided in the Appendix. *We note that solving this problem in an unrolled graph setting was not possible*: either because it was impossible to facilitate learning when a high number of inner-loop iterations was applied (presumably due to the excessive total graph depth), or because of memory issues. Thus, eliminating the unrolled graph is absolutely critical in this case.

## 5 CONCLUSIONS

In this paper we presented NOVAS, an optimization module for differentiable, non-convex inner-loop optimization that can be incorporated in end-to-end learning architectures involving nested optimizations. We demonstrated its advantages over alternative algorithms such as unrolled gradient decent and DCEM on a SPEN benchmark. We also showed that NOVAS allows us to expand the class of PDEs that can be addressed by the deep FBSDE method while resisting the curse of dimensionality, as demonstrated by the solution of a 101-dimensional HJB PDE associated with a portfolio optimization problem. NOVAS is a *general purpose* differentiable non-convex optimization approach and thus, owing to its broad description and its generality, could be useful in a plethora of other applications involving nested optimization operations. We hope that the results of this work will inspire continued investigation.

### ACKNOWLEDGMENTS

This research was supported by AWS Machine Learning Research Awards and NASA Langley. We would also like to thank Brandon Amos for sharing his code for the example of Section 4.1 with us, which allowed us to reproduce his results and benchmark all algorithms.

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

# A APPENDIX

## A.1 NOVAS: DERIVATION AND IMPLEMENTATION DETAILS

Given the optimization problem defined in (3) we calculate the gradient with respect to the sampling distribution parameter $\rho$

$$\nabla_\rho \ln \int S(F(x)) f(x; \rho) \mathrm{d}x = \frac{\int S(F(x)) \nabla_\rho f(x; \rho) \mathrm{d}x}{\int S(F(x)) f(x; \rho) \mathrm{d}x}, \tag{5}$$

$$= \frac{\int S(F(x)) \nabla_\rho \ln f(x; \rho) f(x; \rho) \mathrm{d}x}{\int S(F(x)) f(x; \rho) \mathrm{d}x}, \tag{6}$$

$$= \frac{\mathbb{E}[S(F(x)) \nabla_\rho \ln f(x; \rho)]}{\mathbb{E}[S(F(x))]}, \tag{7}$$

where the second equality is obtained using the log trick. The exponential family distribution is characterized by the probability distribution function

$$f(x; \rho) = h(x) \exp(\rho^{\mathrm{T}} T(x) - A(\rho)), \tag{8}$$

where $h(x)$ is the base measure whose formula depends on the particular choice of distribution, $T(x)$ is the vector of sufficient statistics and $A(\rho) = \ln\{\int h(x) \exp(\rho^{\mathrm{T}} T(x)) \mathrm{d}x\}$ is the log partition. The gradient with respect to log distribution can then be calculated as

$$\nabla_\rho \ln f(x; \rho) = T(x) - \nabla_\rho A(\rho). \tag{9}$$

A Gaussian $\mathcal{N}(\mu, \Sigma)$ is fully defined by its mean and covariance, but one can choose to optimize (3) only over the mean $\mu$, and sample using a fixed covariance. This results in the following parameters of the distribution:

$$T(x) = \Sigma^{-1/2} x, \qquad \nabla_\rho A(\rho) = \Sigma^{-1/2} \mu, \tag{10}$$

and for $\rho = \mu$ the gradient can be calculated as

$$\nabla_\mu \ln \mathbb{E}[S(F(x)) f(\mu)] = \frac{\mathbb{E}[S(F(x))(x - \mu)]}{\mathbb{E}[S(F(x))]}. \tag{11}$$

One reason for choosing to optimize over the mean only is the simplicity of the resulting update expression, as well as numerical reasons, since applying gradient descent on the covariance matrix can lead to non-positive definiteness if the initial values or the learning rate are not chosen carefully. An intermediate solution between using a fixed covariance matrix and its gradient descent-based update law is assuming a diagonal covariance matrix and calculating each element of the diagonal via a simple weighted empirical estimator. The resulting update scheme is given in Alg. 1. Note that we also investigated using the full gradient descent-based update rule for the covariance, as well as the use of the Hessian of the objective function (3) and other techniques like momentum, line search, trainable optimization hyper-parameter values etc. to speed up convergence, but the results were inconclusive as to their additional benefit. We tested two different shape functions: $S(y; \kappa) = \exp(\kappa y)$, as well as a shape function suggested by Zhou & Hu (2014), namely $S(y; \kappa, \gamma) = (y - y_{\min})/(1 + \exp(-\kappa(y - \gamma)))$, where $y_{\min}$ is the minimum of the sampled values and $\gamma$ is the $n$-th biggest value of the assorted $y$ values. For the first choice it is numerically advantageous to include the normalization step in the function definition and replace the exponential with the `softmax` function. The latter choice is a differentiable function approximating the level/indicator function used in CEM. Though both shape functions exhibited similar performance, we noticed that the former was slightly faster, and the latter lead to slightly more accurate results in the SPEN example (but not in the FBSDE example, where they were equivalent). All parameter values such as $\kappa$ can be made trainable parameters of the network, though we did not notice an improvement in doing so. In fact, the algorithm seems to be quite insensitive to its parameter values including the learning rate $\alpha$, with the sole exception of $\sigma$ which does indeed affect its output significantly.

## A.2 STOCHASTIC OPTIMAL CONTROL USING FBSDES

In this section we introduce the deep FBSDE framework for solving PDEs and show that combining NOVAS with the deep FBSDE allows us to extend the capabilities of the latter framework

(Pereira et al., 2019b;a; Wang et al., 2019) in addressing Stochastic Optimal Control (SOC) problems. The mathematical formulation of a SOC problem leads to a nonlinear PDE, the Hamilton-Jacobi-Bellman PDE. This motivates algorithmic development for stochastic control that combine elements of PDE theory with deep learning. Recent encouraging results (Han et al., 2018; Raissi, 2018) in solving nonlinear PDEs within the deep learning community illustrate the scalability and numerical efficiency of neural networks. The transition from a PDE formulation to a trainable neural network is done via the concept of a *system of Forward-Backward Stochastic Differential Equations* (FBSDEs). Specifically, certain PDE solutions are linked to solutions of FBSDEs, which are the stochastic equivalent of a two-point boundary value problem and can be solved using a suitably defined neural network architecture. This is known in the literature as the *deep FBSDE* approach. In what follows, we will first define the SOC problem and present the corresponding HJB PDE, as well as its associated system of FBSDEs. The FBSDEs are then discretized over time and solved on a neural network graph.

Consider a SOC problem with the goal of minimizing an expected cost functional subject to dynamics:

$$\inf_{u \in \mathcal{U}[0,T]} J(u) = \inf_{u \in \mathcal{U}[0,T]} \mathbb{E}\left[\phi(x(T)) + \int_0^T l(x(t), u(t)) \, \mathrm{d}t\right], \tag{12}$$

$$\text{s.t.} \quad \mathrm{d}x(t) = f(x(t), u(t)) \, \mathrm{d}t + \Sigma(x(t), u(t)) \, \mathrm{d}w(t), \quad x(0) = \xi, \tag{13}$$

where $x \in \mathbb{R}^n$ and $u \in \mathbb{R}^m$ are the state and control vectors respectively, $f : \mathbb{R}^n \times \mathbb{R}^m \to \mathbb{R}^n$ is a non-linear vector-valued drift function, $\Sigma : \mathbb{R}^n \times \mathbb{R}^m \to \mathbb{R}^{n \times v}$ is the diffusion matrix, $w \in \mathbb{R}^v$ is vector of mutually independent Brownian motions, $\mathcal{U}$ is the set of all admissible controls and $l : \mathbb{R}^n \times \mathbb{R}^m \to \mathbb{R}$ and $\phi : \mathbb{R}^n \to \mathbb{R}$ are the running and terminal cost functions respectively. Equation (13) is a controlled Itô drift-diffusion stochastic process.

Through the value function definition $V(x,t) = \inf_{u \in \mathcal{U}[0,T]} J(u)|_{x_0=x, t_0=t}$ and using Bellman's principle of optimality, one can derive the Hamilton Jacobi Bellman PDE, given by

$$V_t + \inf_{u \in \mathcal{U}[0,T]} \left[\frac{1}{2}\mathrm{tr}(V_{xx}\Sigma\Sigma^T) + V_x^T f(x, u) + l(x, u)\right] = 0, \quad V(x, T) = \phi(x), \tag{14}$$

where we drop explicit time dependencies for brevity, and use subscripts to indicate partial derivatives with respect to time and the state vector. The term inside the infimum operation is called the *Hamiltonian*:

$$\mathcal{H}(x, u, V_x, V_{xx}\Sigma\Sigma^T) \triangleq \frac{1}{2}\mathrm{tr}(V_{xx}\Sigma\Sigma^T) + V_x^T f(x, u) + l(x, u). \tag{15}$$

Given that a solution $u^*$ to the minimization of $\mathcal{H}$ exists, the unique solution of (14) corresponds by virtue of the non-linear Feynman-Kac lemma (see for example Pardoux & Peng (1990)) to the following system of FBSDEs:

$$x(t) = \xi + \int_0^t f(x(t), u^*(t)) \, \mathrm{d}t + \int_0^t \Sigma(x(t), u^*(t)) \, \mathrm{d}w_t, \quad \text{(FSDE)} \tag{16}$$

$$V(x(t), t) = \phi(x(T)) + \int_t^T l(x(t), u^*(t)) \, \mathrm{d}t - \int_t^T V_x^T(x(t), t)\Sigma(x(t), u^*(t), t) \, \mathrm{d}w, \quad \text{(BSDE)} \tag{17}$$

$$u^*(t) = \arg\min_u \mathcal{H}(x(t), u, V_x(x(t), t), V_{xx}(x(t), t)\Sigma(x(t), u)\Sigma(x(t), u)^T). \tag{18}$$

Here, $V(x(t), t)$ denotes an evaluation of $V(x, t)$ along a path of $x(t)$, thus $V(x(t), t)$ is a stochastic process (and similarly for $V_x(x(t), t)$ and $V_{xx}(x(t), t)$). Note that $x(t)$ evolves forward in time (due to its initial condition $x(0) = \xi$), whereas $V(x(t), t)$ evolves backwards in time, due to its terminal condition $\phi(x(T))$, thus leading to a system that is similar to a two-point boundary value problem. While we can easily simulate a forward process by sampling noise and then performing Euler integration, a simple backward integration of $V(x(t), t)$ would result in it depending explicitly on future values of noise, which is not desirable for a non-anticipating process, i.e., a process that does not exploit knowledge on future noise values. Two remedies exist to mitigate this problem: either back-propagate the conditional expectation of $V(x(t), t)$ (e.g., as in Exarchos & Theodorou (2018)), or forward-propagate $V(x(t), t)$ starting from an initial condition guess, compare its terminal value $V(x(T), T)$ to the terminal condition, and adjust the initial condition accordingly so that

the terminal condition is satisfied approximately. For this forward evolution of the BSDE, the above system is discretized in time as follows:

$$x_{k+1} = x_k + f(x_k, u_k^*)\Delta t + \Sigma(x_k, u_k^*)\Delta w_k, \qquad x_0 = \xi, \qquad \text{(FSDE)} \tag{19}$$

$$V_{k+1} = V_k - l(x_k, u_k^*)\Delta t + V_{x,k}^T \Sigma(x_k, u_k^*)\,\Delta w_k, \qquad V_0 = \psi, \qquad \text{(BSDE)} \tag{20}$$

$$u_k^* = \arg\min_u \mathcal{H}\big(x_k, u, V_{x,k}, V_{xx,k}\Sigma(x_k, u)\Sigma(x_k, u)^T\big). \tag{21}$$

Here, $\Delta w_k$ is drawn from $\mathcal{N}(0, \Delta t)$ and $\mathcal{H}$ is given by eq. (15). Note that for every sampled trajectory $\{x_k\}_{k=1}^K$ there is a corresponding trajectory $\{V_k\}_{k=1}^K$. Under the deep FBSDE controller framework, $V_0 = \psi$ and $V_{x,0}$ are set to be trainable parameters of a deep neural network that approximates $V_x\big(x(t), t\big)$ at every time step under forward-propagation, using an LSTM[3]. The terminal value of the propagated $V\big(x(t), t\big)$, namely $V(x(T), T)$, is then compared to $\phi\big(x(T)\big)$ to compute a loss function to train the network. Note that since the Hamiltonian can have any arbitrary non-linear dependence on the control, the resulting minimization problem (21) is generally non-covex and does not have a closed-form solution. Furthermore, it must be solved for each time step, and for utilization within the deep FBSDE controller framework, the non-convex optimizer must be differentiable to facilitate end-to-end learning. This makes NOVAS a good fit. The neural network architecture is shown in Fig. 3. Since the non-convex Hamiltonian minimization procedure is performed at every time step leading to a repeated use of NOVAS in the architecture, the ability to avoid unrolling the inner-loop computation graph is crucial.

## A.3 FBSDE STOCHASTIC OPTIMAL CONTROL FOR AFFINE-QUADRATIC SYSTEMS

We now show how the previous state-of-the-art (Exarchos & Theodorou, 2018; Pereira et al., 2019b) deals with the problem of the Hamiltonian $\min$ operator by assuming a special structure of the problem. Specifically, they restrict the dynamics of eq. (13) to be affine in control, i.e., of the form $f(x, u) = F(x) + G(x)u$, and the cost in eq. (12) to be quadratic in control, i.e., $l(x, u) = q(x) + u^T R u$. In this case, and if $\Sigma(x, t)$ is not a function of $u$, one can perform explicit minimization of the Hamiltonian with respect to $u$ in eq. (21) to find the optimal control:

$$u^* = -R^{-1}G^T V_x. \tag{22}$$

This is done by simply setting $\partial \mathcal{H}/\partial u = 0$ and solving for $u$. Substituted back into the HJB PDE, this yields a simplified expression without a $\min$ operator:

$$V_t + \frac{1}{2}\text{tr}(V_{xx}\Sigma\Sigma^T) + V_x^T F + q - \frac{1}{2}V_x^T G R^{-1} G^T V_x = 0, \qquad V(x, T) = \phi(x).$$

Thus, for this restricted class of systems, the deep FBSDE neural neural network architecture does not require a numerical minimization operation over $u$ at every time step, as in eq. (21). The cart-pole swing-up task of the next section is an example of a system that satisfies these restrictions. A similar closed-form solution exists for some cases of $L^1$-optimal control (Exarchos et al., 2018), as well as some differential games (Exarchos et al., 2019). While simplifying the problem significantly, this approach comes with an important caveat: several dynamical systems do not have a control-affine structure, and penalizing control energy ($u^T R u$) is not always meaningful in every setting.

### A.3.1 CART-POLE SWING-UP PROBLEM

We define the state vector to be $X = \big[x, \theta, \dot{x}, \dot{\theta}\big]^T$, where $x$ represents the cart-position, $\theta$ represents the pendulum angular-position, $\dot{x}$ represents the cart-velocity, and $\dot{\theta}$ represents the pendulum angular-velocity. Let $u \in \mathbb{R}$ be the control force applied to the cart. The deterministic equations of motion for the cart-pole system are,

$$\ddot{x} = \frac{u + m_p \sin\theta(l\dot{\theta} + g\cos\theta)}{m_c + m_p \sin\theta}$$

$$\ddot{\theta} = \frac{-u\cos\theta - m_p l\dot{\theta}\cos\theta\sin\theta}{l(m_c + m_p \sin\theta)}$$

---

[3]In this work, we additionally use the same LSTM to predict a column of the Hessian $V_{xx}\big(x(t), t\big)$.

For our experiments, we consider the case where noise enters the velocity channels of the state. The stochastic dynamics therefore take the following form,

$$
dX = d \begin{bmatrix} x \\ \theta \\ \dot{x} \\ \dot{\theta} \end{bmatrix} = \begin{bmatrix} \dot{x} \\ \dot{\theta} \\ \dfrac{m_p \sin\theta(l\dot{\theta} + g\cos\theta)}{m_c + m_p\sin\theta} \\ \dfrac{-m_p l\dot{\theta}\cos\theta\sin\theta}{l(m_c + m_p\sin\theta)} \end{bmatrix} dt + \begin{bmatrix} 0 \\ 0 \\ \dfrac{1}{m_c + m_p\sin\theta} \\ \dfrac{-\cos\theta}{l(m_c + m_p\sin\theta)} \end{bmatrix} u\, dt + \begin{bmatrix} 0 & 0 \\ 0 & 0 \\ \tilde{\sigma} & 0 \\ 0 & \tilde{\sigma} \end{bmatrix} \begin{bmatrix} dw_1 \\ dw_2 \end{bmatrix}
$$

The task is to perform a swing-up i.e. starting from an initial state of $X = \begin{bmatrix} 0,0,0,0 \end{bmatrix}^{\mathrm{T}}$ at time $t_0 = 0$, reach the target state of $X = \begin{bmatrix} 0,\pi,0,0 \end{bmatrix}^{\mathrm{T}}$ by the end of the time horizon $t = T$. We consider $T = 1.5$s with a time discretization step of $\Delta t = 0.02$s. Notice that the dynamics are affine in control, and selecting the running cost to be $l = u^T R u$, minimization of the Hamiltonian with respect to $u$ assumes a closed-form solution, namely that of eq. (22). This fact allows us to replace the min operator in favor of this solution (Pereira et al., 2019b). Here, we test NOVAS by avoiding this replacement. We consider a running and terminal cost matrix of $\mathrm{diag}(Q) = [0.0, 10.0, 3.0, 0.5]$ and the control cost matrix of $R = 0.1$. The cart-pole parameters considered are $m_p = 0.01\,kg$, $m_c = 1.0\,kg$, $l = 0.5\,m$, which are the mass of the pendulum, mass of cart, and length of the pendulum, respectively. For the noise standard deviation, $\tilde{\sigma} = 0.5$ was used. As far as the hyper-parameters for learning the deep FBSDE controller are concerned, we used a two-layer LSTM network as shown in Fig. 5(e) with hidden dimension of 16 in each layer, a batch size of 128, and trained the network using the Adam optimizer for 3500 iterations with a learning rate of $5e^{-3}$. For the NOVAS layer at every time step, we used 5 inner-loop iterations and 100 samples for both training and inference. A shape function of $S = \exp(\cdot)$, initial $\mu = 0$, and initial $\sigma = 10$ were used.

With reference to parameters in Alg. 1, for this experiment we used 75 time steps, which means that the LSTM graph can be viewed as a 75 layered feed-forward network when unrolled. Additionally, at each time step we use a **NOVAS_Layer** to compute the optimal control. Thus, the total number of network layers is $L = 75 + 74 = 149$ with $f_i$'s being **NOVAS_Layer** for $i = 2, 4, 6, \ldots$.

### A.3.2 PORTFOLIO OPTIMIZATION PROBLEM

We now consider a problem for which an explicit solution of the Hamiltonian min operator does not exist. Let $N$ be the total number of stocks that make up an index $I$ such that $I = \frac{1}{N}\sum_{i=1}^{N} S_i$, where $S_i$ is the stock price process of the $i$-th stock. Let $M$ be the number of a fixed selection of traded stocks taken from those $N$ stocks such that $M < N$. Furthermore, let $u \in \mathbb{R}^{M+1}$ be the control vector. The $(N+1)$ dimensional state vector is comprised $N$ stock prices and a wealth process $W$. The dynamics of each stock price and wealth process are given by

$$
\pi_k = \big[\mathrm{softmax}(u)\big]_k = \frac{e^{u_k}}{\sum_{m=1}^{M+1} e^{u_m}}, \quad (k = 1, 2, \cdots, M+1) \tag{23}
$$

$$
\mathrm{d}S_i(t) = S_i(t)\,\mu_i\,\mathrm{d}t + S_i(t)\,\mathrm{d}\eta_i \quad \left( \text{where, } i = 1, 2, \cdots, N \text{ and } \mathrm{d}\eta_i = \sum_{j=1}^{N} \sigma_{i,j}\,\mathrm{d}w_j(t) \right) \tag{24}
$$

$$
\mathrm{d}W(t) = W(t)\left( \pi_1\, r\,\mathrm{d}t + \sum_{m=2}^{M+1} \pi_m\,\mu_m\,\mathrm{d}t + \sum_{m=2}^{M+1} \pi_m\,\mathrm{d}\eta_m \right) \tag{25}
$$

where $\pi_k$ is the fraction of wealth invested in the $k$-th traded stock, $r$ is rate of return per period of the risk-free asset, $\mu_i$ is the rate of return of the $i^{\text{th}}$ stock. Here, $\sigma_{i,j}$ denotes the standard deviation of noise terms entering the $i$−the stock process wherein $i = j$ indicates the contribution of the process' own noise as opposed to $i \neq j$, which indicates the interaction of noises between stocks (correlation). All $w_i$'s are mutually independent standard Brownian motions. To obtain the $\sigma$'s, we used randomly generated synthetic covariance matrices which mimic real stock-market data. Note that the $M$ traded stocks were randomly picked and were not constrained to be any specific sub-selection of the $N$ stocks. Separate noise realizations were used during training, validation, and testing to ensure that the network does not over-fit to a particular noise profile.

For our experiments we use $N = 100$ stocks that make up the index $I$ and $M = 20$ traded stocks. We used a scaled squared-softplus function as terminal cost, given by

$$\phi(x(T)) = q\left(\frac{1}{\beta} \cdot \log\left(1 + e^{\beta(I(T) - W(T))}\right)\right)^2,$$

with $\beta = 10$, $q = 500$, and no running cost, focusing on investment outperformance at the end of the planning horizon of one year. To simulate the stock dynamics we used a time discretization of $dt = 1/52$, which amounts to controls (and thus amounts invested) being applied on a weekly basis, for a total time of 1 year. The deep FBSDE-NOVAS hyperparameters were as follows: 16 neurons each in a two-layer LSTM network to predict the gradient of the value function at each time step, a batch size of 32, an initial learning rate set to $1e^{-2}$ and reduced by factor of 0.1 after 4000 and 4500 training iterations. Training was done using the Adam optimizer for a total of 5000 iterations. For NOVAS, we used 100 samples with 5 inner-loop iterations for training and 200 samples with 50 inner-loop iterations for inference. The shape function used was $S(x) = \exp(x)$.

With reference to parameters in Alg. 1, for this experiment we used 52 time steps, which means that the LSTM graph can be viewed as a 52 layered feed-forward network when unrolled. Additionally, at each time step (except for the last time step) we use a **NOVAS_Layer** to compute the optimal control. Thus, the total number of network layers is $L = 52 + 51 = 103$ with $f_i$'s being **NOVAS_Layer** for $i = 2, 4, 6, \ldots$.

### A.3.3 LOSS FUNCTION FOR TRAINING DEEP FBSDE CONTROLLERS

The loss function used in our experiments to train the deep FBSDE controller with the NOVAS layer is as follows:

$$\mathcal{L} = l_1 \cdot H_\delta\big(V(x_T, T) - V^*(x_T, T)\big) + l_2 \cdot H_\delta\big(V_x(x_T, T) - V_x^*(x_T, T)\big)$$
$$+ l_3 \cdot H_\delta\big(V_{xx}(x_T, T) - V_{xx}^*(x_T, T)\big) + l_4 \cdot \big(V^*(x_T, T)\big)^2 + l_5 \cdot \big(V_x^*(x_T, T)\big)^2 + l_6 \cdot \big(V_{xx}^*(x_T, T)\big)^2,$$

where

$$H_\delta(a) = \begin{cases} a^2, & \text{for } |a| < \delta, \\ \delta(2|a| - \delta), & \text{otherwise.} \end{cases}$$

Here, $x_T$ denotes $x(T)$, $V(x_T, T)$, $V_x(x_T, T)$, and $V_{xx}(x_T, T)$ are the predicted value function, its predicted gradient, and the predicted last column of the Hessian matrix, respectively, at the terminal time step. The corresponding targets are obtained through the given terminal cost function $\phi\big(x(T)\big)$ so that $V^*(x_T, T) = \phi(x_T)$, $V_x(x_T, T) = \phi_x(x_T)$ and $V_{xx}(x_T, T) = \phi_{xx}(x_T)$. Each term is computed by averaging across the batch samples. Additionally, we may choose to add terms that directly minimize the targets. This is possible because gradients flow through the dynamics functions and therefore the weights of the LSTM can influence what the terminal state $x(T)$ will be.

For the cart-pole problem we used $\delta = 50$ and $[l_1, l_2, l_3, l_4, l_5, l_6] = [1, 1, 0, 1, 1, 0]$, and for the portfolio optimization problem we used $\delta = 50$ and $[l_1, l_2, l_3, l_4, l_5, l_6] = [1, 1, 1, 1, 0, 0]$.

### A.3.4 HARDWARE CONFIGURATION AND RUN-TIMES

All experiments were run on a NVIDIA GeForce RTX 2080Ti graphics card with 12GB memory. The PyTorch (Paszke et al., 2019) implementation of the 101-dimensional portfolio optimization problem had a run-time of 2.5 hours.

