# OpenReview forum: "NOVAS: Non-convex Optimization via Adaptive Stochastic Search for End-to-end Learning and Control"
_ICLR.cc/2021/Conference — ICLR 2021 Poster_

### Official Review · AnonReviewer2 · 2020-10-28
**Can be improved by a more compelling analysis and principled reasoning for the approach**

**Rating:** 4
**Confidence:** 2

**Review:**

The authors suggest a way to backpropagate through neural network with an embedded optimization problem.
They propose to perform the estimation of the gradient of the optimum of the embedded problem with respect to its parameterization by the differentiation of one step of a stochastic search algorithm.
Their suggested NOVAS approach together with the deep FBSDE method allows for solving optimal control problems efficiently and with low memory due to not unrolling multiple optimizer steps during the backpropagation.
The authors demonstrate the feasibility of their approach via an example with a structured prediction energy network and two optimal control tasks (cart-pole swing-up and portfolio selection).

I do not recommend to accept the paper.

The results are interesting but require a more compelling presentation and in depth analysis.
At the current state it is hard for me to understand how their approach for differentiation of an embedded optimization problem is principled and why it should be better than other approaches.
It is also hard to evaluate whether the presented approach really is better from the given computational experiments.

For a locally strictly convex optimization problem of the form

x^star = argmin_x f(x, theta)

the computation of the Jacobian dx^star / dtheta analytically requires the inversion of the full Hessian matrix via the implicit function theorem:

[partial x^star / partial theta] = -[partial (df/dx) / partial x]^{-1} [partial (df/dx) / partial theta]

where [partial (df/dx) / partial x] is the Hessian of the optimization problem at the optimum.

If x \in R^n then in the locally strictly convex case the Hessian has full rank n.

Backpropagation through only one gradient step contains only information of rank 1.
So the backpropagation of one gradient step generally only contains a small part of the sensitivity of the optimum with respect to theta.

When starting the optimization from different points and repeating throughout multiple outer iterations we potentially gather enough sensitivity information to properly represent the shape of the loss function around the optimum in expectation.
But the approach certainly adds variance compared to having an exact gradient or unrolling many optimization steps.

Additionally, the authors suggest a stochastic search method that due to using the log function trick has low sample efficiency compared to using actual gradient information.
The stochastic search method proposed by the authors gains information about the loss function shape by sampling locally in the input space.
These types of gradient estimators have much higher variance for truly high dimensional loss functions due to the curse of dimensionality.

See Ben Rechts criticism of the policy gradient / REINFORCE gradient estimator that has the same issue:
https://www.argmin.net/2018/02/20/reinforce/

So backpropagation of one gradient step with a high variance gradient estimate compared to the true derivative which has rank n curvature information should be inefficient in theory for truly high-dimensional optimization problems.

Perhaps the authors could add at least some analysis of the variance of their gradient estimator compared to other gradient estimators for embedded optimization problems for different examples and show how the variance behaves depending on the difficulty of the optimization task (dimensionality, curvature) taking the above perspective into account.

Since the authors explicitly treat nonconvex optimization problems they could also make more explicit that a nonconvex optimization problem does not necessarily have a unique optimum.
The argmin is therefore a set-valued map and not differentiable in the classical sense (even when the optimum is unique almost everywhere in the parameter space it is possible that the optimal value is discontinuous in the parameter).
For non-global optimizers (such as gradient descent) starting in such a way that we do not end up in the wrong local optimum is crucial.

It seems problematic that the comparison against the backpropagation through unrolled gradient descent is with an example where the latter finds a different (wrong) local optimum then the suggested NOVAS method.
I don't see how the comparison results with respect to convergence, computational time and inference can be considered conclusive without making the tuning effort for the baseline to perform the desired task properly.

The authors refer to computation time as "inner-loop convergence rate" (Page 6) but convergence rate cannot be inferred from computation time if e.g. sampling gradients using the likelihood ratio trick is much faster than computing a backpropagation gradient.
Reverse mode autodiff gradients are only faster than finite difference / forward mode / sampling gradients for a non-zero constant number threshold of input dimension (how many depends on the efficiency of the autodiff module but it can be e.g. 100-200 input parameters).
The computation time for truly high-dimensional problems can thus look very different than medium dimensional problems (n = 100).

Regarding the violin plots for the portfolio optimization I am not sure how much they benefit the paper.
The goal of the paper should be to demonstrate the superiority of the suggested method over baselines.
The violin plots are relevant to people who are interested in the outcome of that specific optimal control task.
Someone not knowledgeable about the control problem will not benefit much from knowing the terminal wealth of the different strategies.

More analysis about the quality of the gradients obtained by the suggested method compared to other methods would improve the paper:
- How many unrolling steps do what to the gradient variance?
- What is the difference of using backprop through gradients of the loss vs stochastic search gradients (log likelihood trick) gradients for different dimensions of the optimization variable?

Another small suggestions:
On Page 4 "desirable properties of optimization" seems rather vague, perhaps it should be stated more specifically (most likely what is meant is smoothness of the resulting objective function).

---

> ### Author Response · Authors · 2020-11-23
> **Author Response to AnonReviewer2 - Part I**
>
> We thank you for the thorough review and constructive criticism. We really appreciate the time invested to provide both. The main issues raised by the reviewer are:
> 1). Back-propagating through only the last step provides incomplete information about the loss landscape, as the information contained is of rank 1, and information is localized at x*.
> 2). The method we are using has low sample efficiency and, similar to policy gradient methods with which it shares some similarities, will not work for high-dimensional systems.
> 3). Non-convex problems don’t necessarily have a unique optimum, and thus non-global methods (including our own) do not offer guarantees against getting stuck in local optima. The same applies for GD, which the author offers as an explanation as to why it does not perform well in the SPEN example, and calls for further tuning effort in the GD case.
> 4). Computation time is not the same as inner-loop convergence rate (page 6).
> 5). The violin plots offered are not informative by themselves as a metric for comparison between the presented baselines.
> 6). Improvement suggestions.
>
> In a point-by-point response:
>
> 1). Without wanting to question the argumentation, we have to admit that we struggle to understand why a single step x’ → x* contains only information of rank 1 whereas a trajectory of several iterations x1 → x2 → … → x* is higher rank. This is most likely a confusion from our side, but wouldn’t several iterations perform successive multiplications of the Hessian? How could that increase the rank? In any case, if one does not unroll the graph for GD, then the information at the last step would indeed seem to be localized around x*. But for sampling-based optimization, x* is by definition nothing more than a convex combination of the N sampled x’s: x* = Sum_i w_i x_i, for non-negative w_i that sum up to 1 (in the more general case, there can also be momentum w.r.t. the value of the previous iteration). Doesn’t it seem a bit odd that information is in this case still exclusively localized at a single value, x*?
>
> 2). This is a very interesting point. Again, without wanting to question the Reviewer’s statements, we would like to argue that sampling-based optimization is a well-established field with lots of documented success. CEM, CMA-ES, Monte Carlo sampling and countless gradient-free variants have been used for many years, with notable results. Concerning policy gradient, we agree that its vanilla REINFORCE version is hardly a good option, but policy gradient algorithms such as TRPO and PPO have been dominating the field of robotics for the past 5 years and work very well in high-dimensional tasks. Furthermore, Adaptive Stochastic Search has been used as a trajectory optimization method (e.g. “Constrained sampling-based trajectory optimization using stochastic approximation”, Boutselis et al.) and so has CEM (see “Sample-efficient CEM for real-time planning”, Pinneri et al.) in which a discretized (in time) control sequence is treated as the optimization variable and is optimized in its full dimensionality. The resulting control update law in the case of the former is a generalization of the Path Integral control law (see for example MPPI) as well as Information Theoretic MPC. In the controls field, all these algorithms have been used extensively and demonstrated solid performance. Rather than disagreeing with the Reviewer, we are merely surprised by this performance assessment.
>
> 3). We updated the manuscript to include the Reviewer’s insight on the argmin being potentially a set-valued map and not differentiable in the classical sense. We agree with the Reviewer concerning the vulnerability of both GD and sampling-based optimizers to local optima. There is some evidence that Gradient-based optimization is actually more prone to getting stuck in local optima, since sampling-based approaches evaluate the objective function over an extended area of the input space and, given enough sampling variability for exploration, have thus more chances of escaping a narrow local optimum. This seems to be the case in our SPEN example as well: all optimizers were initialized at the same x = 0; only unrolled GD failed to obtain a good solution. We did experiment with GD hyper-parameters but did not observe significant difference. We feel that changing the GD initialization would provide an unfair advantage not just to our method, but also to unrolled DCEM. Note that our work is not the first to report such failure of unrolled GD; the authors of DCEM made a similar observation.
>
> 4). We totally agree. Our remark on inner-loop convergence is based on Fig. 2 (c), not (b) reporting computation time. 2(c) compares only DCEM with NOVAS, since nothing can be inferred about GD from that figure.

---

> > ### Author Response · Authors · 2020-11-23
> > **Author Response to AnonReviewer2 - Part II**
> >
> > 5). We agree that the violin plots of Figure 5(b) are informative only to people who are interested in the outcome of the specific optimal control task (performance compared to the wealth index). This plot was indeed provided for those curious about the actual application, and to show that the end result is reasonable (approximately 5% over the market average is a realistic performance gap). However, the violin plots of Figure 5(a) represent the total cost (i.e., the performance objective of the optimal control problem) of our method compared to the baselines. This does not require any knowledge about the particular problem and does indeed show superiority over the baselines. Since the outcome is highly stochastic and volatile (the noise being a dominant factor), rather than presenting the mean cost, we present the approximate full distributions. We updated the captions for clarity. Would the Reviewer prefer a different presentation?
> >
> > 6). We agree that more investigation and analysis on hyper-parameters would be beneficial, and this on several different applications, including meta-learning as other reviewers suggested. We feel that such a broad-usage module as the one we propose would need to be evaluated separately in each application domain (deep-FBSDEs, SPENs, meta-learning, and many more), but it would be impossible to do so for all fields within a single publication. In this paper, our emphasis was on non-convex dynamic optimization for large-scale stochastic optimal control problems.  Problems on stochastic dynamic optimization using optimality principles to solve non-convex Hamiltonians are challenging and scalable algorithms for such problems are non-existent. Therein lies the value of our proposed method.
> >
> > We would like to argue that even with the potential weaknesses mentioned by the Reviewer, we believe that this work serves a purpose: (A) it offers an alternative to unrolled GD and unrolled DCEM which researchers and practitioners can test within their own research field. As of now and to the best of our knowledge, only the aforementioned two are available for the type of optimization we consider. (B) compared to unrolled GD, preliminary results show potential decreased vulnerability to local optima, a correctly recovered energy landscape, and no overfitting to hyperparameters (seen in the SPEN example). (C) compared to unrolled DCEM, we offer a 5x speed increase, no need to solve additional convex problems to approximate the eliteness threshold, and higher inner-loop convergence rate. (D) we show that at least in the presented examples, unrolling the graph is not only unnecessary but also comes at a prohibitive memory cost, and (E) we present a new state of the art in deep FBSDE literature by solving fully nonlinear HJB PDEs with non-convex Hamiltonians, which was previously impossible. We believe that these results, despite not being immediately generalizable to every possible domain of application, nevertheless show a clear potential of this method to be useful and even preferable to the aforementioned alternatives. As such, we feel that a publication is justifiable.

---

### Official Review · AnonReviewer4 · 2020-10-28
**Efficient Inner Optimization module for networks**

**Rating:** 6
**Confidence:** 3

**Review:**

This paper proposes using adaptive stochastic search as an optimization module within deep neural networks to perform general non-convex optimization.  This is used as a block within deep FBSDEs,which in general do not have a closed form optimization solution, and use the resulting network to show state of the art results in solving high dimensional PDEs on a 101 dimensional portfolio optimization problem.

Pros :
1. Using adaptive stochastic search allows the inner optimization module to take multiple iterations without unrolling the computation graph (unlike meta-learning methods), since the initial value used by the module is arbitrary, and not provided by the network. This greatly reduces memory requirements, and also speeds up learning and convergence. The authors empirically show a 5x speed-up in using their approach compared to unrolled differentiable cross entropy and better qualitative performance than unrolled gradient descent for training an energy function for a simple regression task.

2. Authors confirm that their approach of using the optimization approach within a deep FBSDE works as expected by getting the optimal solution for cartpole, then use their method to beat random and constant strategies on a high dimensional portfolio optimization problem, for which it is not possible to use methods that unroll the computation graph (due to memory issues).


Cons:
1. More experiments for Structured Energy Prediction Networks with more challenging functions would give a better indication of the limits of the proposed approach in comparison to prior work. It is also unclear if the proposed approach would show worse performance for non-toy datasets where a few iterations of unrolling gradient descent or differentiable cross entropy are sufficient. This is especially since the only non-toy experiment was performed in a domain where other methods (differentiable cross entropy and gradient descent) couldn't be run.

---

> ### Author Response · Authors · 2020-11-23
> **Author Response to AnonReviewer4**
>
> We would like to thank you for the positive reception of our paper. We absolutely agree that with respect to SPENs, the results presented are not enough to claim any advantage or superiority of our method specifically within the SPEN literature. The SPEN toy-example is used merely as a simple benchmark illustrating the various differences between unrolled GD, unrolled DCEM, and our method. More experiments are indeed needed to establish a proof of superior performance of our algorithm in the SPEN domain; this is left as a future research direction because of lack of space in the paper, as well as due to the fact that our emphasis in this paper was on non-convex dynamic optimization for large-scale stochastic optimal control problems.  We would have liked to compare the alternatives in the finance example, but their memory cost is prohibitive; we feel that this fact by itself advocates for the utility of our proposed approach.

---

### Official Review · AnonReviewer1 · 2020-10-28
**Great work!**

**Rating:** 6
**Confidence:** 2

**Review:**

### Summary

The authors present the idea of adaptive stochastic search as a building block for neural networks, as an alternative to other "inner loop" optimization methods like gradient descent.

### Reasons for score

Presented works well against the baselines selected and the claims are supported by empirical evidence.

### Pros

1. Paper is clear and grounded in existing literature and context
1. Claims are supported by empirical evidence.
1. Proposed method is robust to changes in hyperparameters like the number of inner loop iterations during inference.
1. Proposed method allows learning the intended functions (see Figure 1) leading to better generalization, the modularity is a bonus.

### Cons

1. There are some cases (although not in general) of meta-learning for adaptation, for example where the update can be described in an implicit fixed point way or when the method used in [FirstOrderMAML](https://lilianweng.github.io/lil-log/2018/11/30/meta-learning.html#first-order-maml) , which allows not needing to unroll (when backpropagating) the function to be optimized. The paper doesn't present this relevant information in page 4. (Edit: this is fixed now)
1. In Figure 2 clarifying if the loss is train set or test set would be nice, plotting both would be even better. (Edit: this is fixed now)
1. Not comparing to existing task-adaptation/meta-learning methods and benchmarks (for example First-Order MAML's gradient)

### Suggestions

1. In Figure 1 both (b) and (c) visibly look really similar, I would suggest plotting the difference between them, so a reader can understand better via visualizing the energy function the advantages/disadvantages of (not) unrolling.  (Edit: the authors considered it)

---

> ### Author Response · Authors · 2020-11-23
> **Author Response to AnonReviewer1**
>
> Thank you very much for the positive reception of our paper. Concerning the Reviewers’ feedback: (1) we were not aware of the results mentioned by the Reviewer in the meta-learning literature that do not involve unrolling of the graph. We included First-Order MAML as well as implicit MAML (which is what we think the Reviewer is also referring to) in the literature review. We thank the Reviewer for pointing out this omission. (2) Thank you for mentioning it: the results depicted in Fig. 2 (a) are losses on the test set. The caption has been updated. (3). We agree that testing NOVAS in meta-learning would be an exciting future research direction; we did not pursue this for lack of space in the paper, and because our emphasis in this paper was on non-convex dynamic optimization for large-scale stochastic optimal control problems. Concerning the suggestion for Fig 1 (b) and (c), we mainly wanted to show that NOVAS, in both its rolled and unrolled form, does indeed recover the correct energy landscape, as opposed to unrolled GD. If we were to plot their difference, this would distort the comparison to the ground-truth landscape.

---

### Official Review · AnonReviewer3 · 2020-11-02
**Official Blind Review #3**

**Rating:** 6
**Confidence:** 2

**Review:**

**Summary**
This paper aims to present a method that allows efficient learning in neural networks architecture that present optimization blocks. These blocks have the form of x_{i+1} = \arg \min_x F(x, x_i, \theta), and can be thought of as a neural network layer. The addition of this block results in a complex optimization problem, since it presents a multi-level problem. The approach presented in this paper relies on adaptive stochastic search as a differentiable optimization procedure. The authors evaluate the proposed algorithm in a variety of applications, including structured prediction networks and control.

**Assessment**
While the paper is well written, its structure could be significantly improved and the problem statement could be made clearer -- since the topic of the paper is not common, I believe further explanation should be made (it could be by the means of a graph or a figure).

*Pros:* The algorithm presented here it’s simple and seems to lead to good results. It is benchmarked in a variety of fields (1) energy-based learning, (2) robotic control, and (3) portfolio management. In all the cases exceeding the presented baselines.


*Cons:* First of all, the paper does not present or mention the practical implementation of their algorithm, which I believe is the only contribution of their work. As it is right now, there is no section of their technical contribution -- just mention of the existing adaptive sampling. Secondly, the baselines introduced, except section 4.1, are non-existent (section 4.2) or extremely naive (section 4.3). Finally, it lacks further experiments underpinning the claims made and ablations.

Here I provide more detailed feedback of my “cons” points. Regarding my first point, since this paper presents a simple-idea/existing-idea-in-another-context, I would expect more details of the practical implementation of their algorithm and sensitivity to the different hyper-parameters. Both of those are lacking in the main paper. The experiment section, while it tackles three different domains, it does not provide enough evidence with respect to previous methods. Section 4.2, depicts a toy reinforcement learning/control problem and no baseline is provided. I would like to (1) incorporate the baseline in Pereira et al. 2019 and LQR with an analysis of the run-time for each algorithm and performance, and (2) test the algorithm to be tested on more complex domains and provide baselines that present the same assumptions, e.g., iLQR. One of the main reasons for the presented method is that it can tackle non-convex objectives; however, the environment presented is convex. Finally, the authors make a lot of emphasis on the difference between their optimization approach and the one that meta-learning does. It seems to me that at least U-NOVA could perform the same task as MAM, or even NOVA with some context of the task. I would like further explanations of this, and some experiments with (U-)NOVA on meta-learning. Finally, I would like further clarifications on why the graph decoupling to get a better initialization is justifiable with NOVA and not other procedures.

---

> ### Author Response · Authors · 2020-11-23
> **Author Response to Official Blind Review #3**
>
> Thank you for your feedback. Following your suggestion, we moved the NOVAS description given by  Alg. 1 from the Appendix to the main body of the paper, along with more details concerning its practical implementation. Several additional details, including hyperparameter sensitivity, are included in the Appendix (Section A1). Concerning the lack of a baseline for the cart-pole example of Section 4.2, we didn’t include one because there is no reason to use NOVAS in such a problem; the Hamiltonian minimization problem treated by the NOVAS module in this particular case is not only convex but also quadratic, and thus allows for a closed-form solution, which is what Pereira et al. exploit to solve it using deep FBSDEs. We included this example merely as a “sanity check”: the cart-pole problem is well-known and has an easily interpretable solution, and we wanted to show that combining NOVAS with deep FBSDEs would yield the usual, expected results. We expect the NOVAS-reliant deep FBSDE to be slower and possibly less accurate in this case than the deep FBSDE algorithm by Pereira et al. since it is numerically calculating a solution that has been “hard-coded” in the latter, but the utility of NOVAS lies in problems that do not allow for such analytic solutions. In the finance example, the Hamiltonian minimization is non-convex and such a closed-form solution does not exist, therefore the framework by Pereira et al cannot be used. Nevertheless, the finance example is less well-known than the cart-pole and with perhaps less intuitive/anticipated solution. We therefore opted to present results for the cart-pole as well, mainly to increase the readers’ confidence in the algorithm. Concerning the use of U-NOVAS in meta-learning, the Reviewer is absolutely right; U-NOVAS can indeed replace the typical unrolled gradient descent-based adaptation rule. However, for lack of space, and because our emphasis was on non-convex dynamic optimization for large-scale stochastic optimal control problems, we did not investigate this. Nevertheless, we did include this observation in the introduction of the updated manuscript. An important final issue is raised by the Reviewer, namely whether other methods such as DCEM are unable to avoid unrolling the graph. We do not make such a claim (in fact, we believe that one could indeed possibly avoid it, though still having to deal with the significant computational overhead of DCEM over NOVAS, as well as its slower convergence rate). However, we implemented DCEM exactly as it was proposed in the cited paper, and the implementation involves unrolling the graph (a fact we also confirmed by communicating with the DCEM lead author directly). The authors of DCEM intended it to be unrolled, and the discovery that unrolling can be unnecessary in such problems is part of the contributions of our paper.

---

### Author Response · Authors · 2020-11-23
**First Revision Author Response**

We would like to thank all reviewers for their constructive feedback and suggestions. We updated the manuscript to reflect the Reviewers' improvement suggestions; changes in the manuscript are indicated by blue font for convenience.

---

### Decision · Program_Chairs · 2021-01-07
**Final Decision**

**Decision:**

Accept (Poster)

**Comment:**

The authors propose an intriguing alternative to IFT or unrolled GD as a method for optimizing through arg min layers in a neural net, by using a differentiable sampling-based optimization approach. I found the general idea in the paper to be intriguing and thought-provoking. The reviewers generally seem to have also appreciated the method, and many of the reviewers' concerns were addressed by the authors during the rebuttal. Although the paper does have a number of flaws -- in particular, the evaluation is a bit hard to appreciate, since improvement over prior work is either unclear, or no meaningful comparison is offered, -- I think in this case the benefits outweigh the downsides. The work is far from perfect, but the ideas that are presented are interested and valuable to the community, and I think that ICLR attendees will appreciate learning about this work. I would encourage the authors however to improve the paper, and especially the empirical evaluation, as much as possible for the camera-ready, and to take reviewer comments into account insofar as feasible. I'm also not sure how much I buy the "overfitting to hyperparameters" argument for unrolled GD, and a less charitable interpretation is that the authors present this issue largely to make up for the comparative lack of other benefits. That's not necessarily a bad thing, but I think making such a big deal of it is a bit strange. It's probably fair to say at this stage that the actual benefits of this approach are a bit modest (though improvements in runtime are a good thing...), but the idea is interesting, and may spur future research.